# OpenReview forum: "Learning from Reference Answers: Versatile Language Model Alignment without Binary Human Preference Data"
_ICLR.cc/2026/Conference — ICLR 2026 Conference Withdrawn Submission_

### Official Review · Reviewer_5LPR · 2025-10-29

**Soundness:** 3
**Presentation:** 2
**Contribution:** 2
**Rating:** 4
**Confidence:** 4

**Summary:**

This work introduces an alignment method that is an alternative to the most common way of performing alignment: using binary preference judgments from humans and training reward models on those judgments for use during RL. Instead of using binary judgments, it uses unary signals. Specifically, it uses the similarity (as measured by traditional similarity metrics such as BERTScore and METEOR) between generations and a reference answer as proxies for reward model signals. The authors find performance similar to existing methods using this approach.

**Strengths:**

1. To my knowledge, the use of existing text similarity metrics as a proxy reward signal is novel. While some methods like KTO (which the authors cite and compare against) don't use preference judgments, this method seems to be more efficient.
2. The experiments are pretty thorough and clearly presented, with extensive comparisons with baselines.
3. The comparisons with different similarity metrics, and especially the analysis with respect to factors such as response length, were very interesting and demonstrate the trade-offs of using different metrics.

**Weaknesses:**

1. My main concern is regarding the scalability of this approach. While it is certainly true that preference annotations are costly to obtain, the same is true for reference answers. High-quality references require much more validation, and that validation itself can be a lot trickier to actually perform. In comparison, relative judgments can be more robust "out-of-the-box." That being said, I think there is certainly an argument for using high-quality references as opposed to binary preferences in certain applications. For instance, I think this may be true for tasks requiring niche domain expertise. I think this work would be more compelling with a more targeted analysis for certain tasks where this is the case, as opposed to the more general helpfulness/harmfulness framework.
2. There has been a lot of work showing the flaws of automatic metrics (e.g. [1]), especially when references are low-quality. To be clear, the authors do address the trade-offs and particular biases of using certain metrics (such as some metrics causing a bias towards longer responses). However, I think these flaws compound the issue described in the previous weakness. While reward hacking is always a concern even with the usual BT-based setup, I worry that this might be even more prevalent and subtle when using these flawed metrics. Again, with this, I think the paper would be stronger with a more targeted analysis over different domains.
3. While I do think the analysis of outputs is helpful in the paper, I think it could be expanded to analyze, for instance, the diversity of the resulting outputs after using RefAlign. I also think a small human validation/comparison experiment would be helpful.

[1] Goyal, Tanya, Junyi Jessy Li, and Greg Durrett. "News summarization and evaluation in the era of gpt-3." arXiv preprint arXiv:2209.12356 (2022).

**Questions:**

1. Why is recall used for BERTScore instead of F1 (or even precision)?
2. For the ~30% of cases where BERTScore does not correspond to the RM, did you notice any patterns?
3. Is there a risk of inducing even more mode collapse when optimizing with respect to one reference? Was this something you noticed when analyzing the outputs after using RefAlign?

---

> ### Author Response · Authors · 2025-11-20
> **Similarity-based Rewards are low-cost alternatives to traditional RMs**
>
> Hi Reviewer 5LPR,
>
> Thank you for your time and feedback. Below, we will address your concerns point by point.
>
> ## 1. My main concern is regarding the scalability of this approach.
> ---
> - **We can directly use the chosen response as the reference. This maintains the scalability of our method to be at least as good as that of traditional RLHF with binary preference data.** *Namely, we can also use relative judgement to choose reference answers, but we do not need the rejected one.* As stated in lines 074-075,
> >"Only the chosen answers in the previous preference data annotation pipeline are required, namely, the unary reference answers. Selecting the rejected ones or constructing large numbers of pairs from multiple candidates for each prompt becomes unnecessary."
>
>   This is also what we did in confidence alignment (Table 3), where we use the chosen response as the reference.
> - We thank the reviewer for the suggestion "for tasks requiring niche domain expertise....a more targeted analysis for certain tasks where this is the case". Actually, we chose safety alignment for this reason. In most cases, safety alignment requires the model to say "NO" to harmful questions. There are certain patterns for good answers, as we show in Appendix C.1.2. For example, "I cannot...". After RL with similarity-based rewards, we found that the aligned model does learn such patterns from the reference. An analysis can be found at lines 352-355. Example responses can be found at Table 7.
>
> ## 2. On the Flaws of Automatic Metrics
> ---
> - It is true that automatic metrics have some flaws. As the reviewer said, even the popular Bradley-Terry reward models suffer from reward hacking.
> - **We believe the key is the trade-off between cost and performance.** **Our work offers an alternative choice for people who want to do certain alignment for LLMs besides the traditional pipeline:** preference data collection, reward modeling, and alignment.
> >"(lines 018~020) When unary reference answers are available, such similarity-based rewards can circumvent the need for binary preference data and explicit reward modeling".
>
>   Our work offers another low-cost choice for alignment.
> - In three alignment cases, including safety, confidence, and general preference alignment, we found that automatic metrics achieve comparable results to popular Bradley-Terry reward models. This shows that automatic metrics are effective alternatives to reward models in common alignment tasks.
>
>
> ## 3. Why is recall used for BERTScore instead of F1 (or even precision)?
> ---
> - At the beginning of this work, we chose recall because it is the best metric among recall, precision, and F1 in most cases in the original paper[1].
> - Using precision as the reward leads to unstable RL optimization processes and even model collapse. As Eq.(13) in the paper shows, one tricky way to increase precision is to produce short responses. A short but bad response may have a larger score than a long and good response.
> - Using Recall is slightly better than using F1-score in most cases. Here we provide a specific example in Table 5LPR-1. Thus, we use recall as our first choice.
> - **Table 5LPR-1.** Mistral-7B-Instruct-v0.2 with SimPO, BertScore as the reward function, Evaluation on AlpacaEval 2 with `gpt-4o-mini` as the referee.
> |    Method    | BertScore Type | LC    | WR    |
> |:------------:|----------------|-------|-------|
> | Ref. + SimPO | F1-Score       | 30.55 | 33.29 |
> | Ref. + SimPO | Recall         | 30.11 | 36.46 |
>
> ## 4. For the ~30% of cases where BERTScore does not correspond to the RM, did you notice any patterns?
> ---
> - In these cases, **the top response ranked by BERTScore always resembles the reference.** Here we provide two examples. Among all 4 candidate sentences, BERTScore always put the one resembles the reference answer as the top rank. The pattern of the BERTScore top ranked sentence is determined by the reference.
> - Please check the two examples in the following comment.
>
>
> ## 5. Is there a risk of inducing even more mode collapse when optimizing with respect to one reference?
> ---
> - We do not notice model collapse when optimizing with respect to one reference. In both confidence alignment and general preference alignment, we optimize the model with respect to one reference, and we do not find model collapse in these cases.
> - For multiple references, we manually adjust the weight for a certain reference as we did in safety alignment. Please refer to Eq.(8), where we introduce a hyper-parameter $\alpha$ to control the importance of harmlessness for safety alignment. By adjusting $\alpha$, we can control the harmlessness of the aligned model. We do not observe model collapse in such multiple reference cases, either.
>
> ## Reference
> ---
> [1] https://arxiv.org/pdf/1904.09675
>
> [2] https://huggingface.co/datasets/Open-Orca/OpenOrca
>
> [3] https://huggingface.co/datasets/CarperAI/openai_summarize_comparisons
>
>
> ---
> Please do let us know if you have any other questions.
>
> Thank You.
>
> Best

---

> > ### Author Response · Authors · 2025-11-20
> > **Examples for previous Point 4**
> >
> > ## 1. Example 1
> > ---
> > - **Source**: OpenOrca [2]
> > - **Model**: Mistral-7B-Instruct-v0.2
> >
> > - **Prompt**: You are a helpful assistant, who always provide explanation. Think like you are answering to a five year old. Write a sentence based on this summary: iraqi embassy in jakarta removes saddam hussein 's photo
> >
> > - **Reference (chosen response in [2])**: The building in Jakarta where people from Iraq work, took down a picture of a man named Saddam Hussein.
> >
> > - **BertScore Top Rank**: .\n\nThe embassy in Jakarta took down a picture of Saddam Hussein because he is no longer the leader of Iraq.
> >
> > - **RM Top Rank:** from its building.\n\nThe Iraqi embassy in Jakarta took down a picture of Saddam Hussein that was hanging on their wall. It's like when you take down a picture of your favorite teddy bear from your bedroom wall. But instead of a teddy bear, it was a picture of a man named Saddam Hussein. And instead of a bedroom wall, it was on the wall of the Iraqi embassy in Jakarta. Why did they take it down? I'm not sure, but maybe they wanted to make their building look different, or they didn't want to remember Saddam Hussein anymore. Either way, it's like changing a picture on your wall at home.
> >
> > ## 2. Example 2
> > ---
> > - **Source**: CarperAI/openai_summarize_comparisons
> > - **Model**: Mistral-7B-Instruct-v0.2
> > - **Prompt**:
> > USER: Summarize the following article as a TL;DR.\nSUBREDDIT: r/relationship_advice\nTITLE: [32/M] My girlfriend [39/F] is very upset that I didn't do enough for her birthday\nPOST: Friday was my girlfriend's birthday. She worked during the daytime and her teenage daughter got her tickets to go to a comedy show all evening which started not too long after she got out of work, so I didn't think I would see her at all on her actual birthday. .....*ignore 1,626 characters*..... :(\nASSISTANT:
> >
> > - **Reference (chosen response in [3])**: TL;DR:  I didn't do anything for my girlfriend's birthday and now she's upset.
> >
> > - **BertScore Top Rank**: TL;DR:  I wasn't excited about my girlfriend's birthday and didn't do anything for it. She is very upset. I feel horrible.
> >
> > - **RM Top Rank**: TL;DR: The boyfriend unexpectedly celebrated his girlfriend's birthday the night before due to her work schedule and her request for a quiet night at home. She was initially happy, but later expressed disappointment and anger for not receiving any gifts or attention on her actual birthday, leaving the boyfriend feeling regretful and unsure how to make it right.
> >
> > ## Reference
> > ---
> > [1] https://arxiv.org/pdf/1904.09675
> >
> > [2] https://huggingface.co/datasets/Open-Orca/OpenOrca
> >
> > [3] https://huggingface.co/datasets/CarperAI/openai_summarize_comparisons

---

> > > ### Comment · Reviewer_5LPR · 2025-11-27
> > >
> > > Thank you for the thorough response. Many of my concerns have been addressed, but I'm still unsure about the response to the first weakness I listed. I may be missing something obvious, but I still don't quite understand how the method is tailored towards reducing cost when considering the response to that first weakness?

---

> > > > ### Author Response · Authors · 2025-11-28
> > > > **Similarity-based Rewards are low-cost alternatives to traditional RMs**
> > > >
> > > > Hi Reviewer 5LPR,
> > > >
> > > > Thanks for your time and response.
> > > >
> > > > ## 1. Conceptual Comparison between traditional RLHF and RefAlign
> > > > ---
> > > > - **Traditional RLHF**: preference data collection ===> reward modeling ===> alignment
> > > > - **RefAlign**: reference data collection ===> alignment
> > > > - As we can also use pair-wise comparison to choose the reference answers, the annotation cost of RefAlign is no more than that of traditional RLHF. Suppose we are annotating $K\ge2$ responses given a prompt in a pair-wise manner, the time for choosing the best one is $\mathcal{O}(K-1)$, which is no more than the time for labeling all pairs $\mathcal{O}((K-1)K/2)$ [1,2].
> > > >     - For relatively small models, it would be more straightforward to get the reference answers; this is one merit of our method.
> > > > - *We do not need the reward modeling process.* This would be great news for cases where multiple RMs are required [3,4].
> > > >
> > > > ## 2. Specific Comparison
> > > > ---
> > > > ### Safety Alignment
> > > > ---
> > > > - In Table 2, SafeRLHF[4] **recruits crowdworkers** to annotate preference data and train a **7B reward model** plus a **7B cost model.** This process goes for three rounds. SafeRLHF then conducts safety alignment with PPO, which includes a  **7B reward model** and **7B cost model.**
> > > > - In Table 2, RefAlign uses chosen response from SafeRLHF[4] or Llama-3.3-70B-Instruct-AWQ as the reference, the BERTScore model is **bart-large-mnli, 407M.** RefAlign avoids the cost of training and alignment with reward models in this case.
> > > >
> > > > ### Confidence Alignment
> > > > ---
> > > > - In Table 3, CONQORD[5] uses Anthropic/hh-rlhf (labeled by humans) to train a **7B or 13B reward model**, then conducts alignment with PPO, which includes these **7B or 13B reward models**.
> > > > - In Table 3, RefAlign uses the chosen response from CONQORD as the reference. The BERTScore model is  **bert-large-uncased,  336M**.
> > > >
> > > > ---
> > > > Overall, we believe that **similarity-based reward functions are low-cost supplements to the traditional reward functions.** It may not be a full replacement for RMS, but a low-cost choice for people in a wide variety of cases.
> > > > >When unary reference answers are available, such similarity-based rewards can circumvent the need for binary preference data and explicit reward modeling.
> > > >
> > > > ---
> > > > [1] OpenAI. Learning to summarize from human feedback. 2022.
> > > >
> > > > [2] OpenAI. Training language models to follow instructions with human feedback. 2022.
> > > >
> > > > [3]  Meta. Llama 2: Open foundation and fine-tuned chat models. 2023.
> > > >
> > > > [4] Dai et al. Safe rlhf: Safe reinforcement learning from human feedback. 2024.
> > > >
> > > > [5] Tao et al. When to trust LLMs: Aligning confidence with response quality. 2024.
> > > >
> > > >
> > > > ---
> > > > Best
> > > >
> > > > Submission7493 Authors

---

### Official Review · Reviewer_Rw8W · 2025-11-01

**Soundness:** 2
**Presentation:** 3
**Contribution:** 2
**Rating:** 2
**Confidence:** 4

**Summary:**

The paper proposes RefAlign, a reinforcement learning (RL)-based alignment method for large language models (LLMs) that leverages similarity (BERTScore) between model-generated responses and high-quality reference answers as a surrogate reward function, eliminating the need for costly binary human preference data. RefAlign simplifies the alignment pipeline and can be effectively adapted for general human preference, safety, and confidence alignment scenarios. Empirical experiments demonstrate that RefAlign achieves comparable performance.

**Strengths:**

1. This paper proposes an approach that leverages similarity between generated responses and high-quality reference answers as a surrogate reward, thereby eliminating the dependency on binary human preference data.
2. The paper thoroughly validates its approach across various tasks and datasets.

**Weaknesses:**

Although the method proposed in this paper addresses the issue of reducing dependency on reward models, it introduces several limitations. First, the requirement of high-quality reference answers imposes constraints, as collecting these unary reference responses still demands significant human effort—albeit potentially less than binary labeling. More fundamentally, a key advantage of traditional reinforcement learning from human feedback (RLHF) is its capability to optimize models solely based on prompts without requiring pre-existing high-quality responses. In contrast, the algorithm presented here necessitates high-quality reference responses, making practical application somewhat limited. Lastly, while this method replaces the reward model with a text-similarity model, the actual reduction in computational and annotation costs appears minimal, as one form of labeling and model dependency is merely substituted for another.
When the quality of model-generated responses is actually higher than the reference answers, similarity-based scoring metrics can have counterproductive effects.

**Questions:**

See Weaknesses.

---

> ### Author Response · Authors · 2025-11-20
> **Position: A Supplement to the Reward Function, Not a Full Replacement for RMs**
>
> Hi Reviewer Rw8W,
>
> Thank you for your time and feedback. Below we will address your concerns point by point.
>
>
> ## 1. First, the requirement of high-quality reference answers imposes constraints, as collecting these unary reference responses still demands significant human effort—albeit potentially less than binary labeling.
> ---
> - As stated in lines 073-075, "Only the chosen answers in the previous preference data annotation pipeline are required", **our method is compatible with the previous annotation pipeline without labeling rejected ones or constructing large numbers of pairs from multiple candidates.** *We can use the chosen answers from the previous preference dataset as reference answers as we did in confidence alignment in Table 3.*
> - It is true that collecting binary preference data or reference answers requires human effort. This is the bottleneck of alignment. It seems that human effort is necessary in the alignment stage, as we want the model to follow human preference, which sources from human efforts. As the reviewer notes, the annotation cost of our method is "potentially less than binary labeling". We think this is one of our contributions, and it is also valuable, especially in cases where multiple candidates are available for a prompt or the reward modeling process is heavily costly.
> - Overall, **similarity-based Rewards are low-cost alternatives to traditional RMs.**
>
> ## 2. a key advantage of traditional RLHF is its capability to optimize models solely based on prompts without requiring pre-existing high-quality responses
> ---
> - It is true that this is a merit of reward models, so we state at lines 017-019
> >"When unary reference answers are available, such similarity-based rewards can circumvent the need for binary preference data and explicit reward modeling".
> - **The key of the question is whether the prompt set used for alignment ($D_{align}$) is a subset of/equal to that used for reward modeling ($D_{RM}$).** **In many cases, $D_{align} \subseteq D_{RM}  $.** For example, Anthropic (Section 4.1 in [1]), LLaMA-2 (Section 3.2.1 in [2]), LLaMA-3 (Section 4/Fig. 7 in [3]), and academic works like SimPO [4] or DPO. *For the baselines of safety alignment and confidence alignment, we also have $D_{align} \subseteq D_{RM}$ in most cases.*
> - In summary, *our method can be an alternative to traditional RMs in a wide range of cases.*
>
> ## 3. replaces the reward model with a text-similarity model, the actual reduction in computational and annotation costs appears minimal
> ---
> - *This is not accurate.*
> - For safety alignment in Table 2 and confidence alignment in Table 3, the reward models in the PPO baselines are roughly around **7B**, while the sizes of BERTScore models are **336M or 407M.** **More than 10x decrease in the parameters for RMs.**
> - **The above comparison does not include the cost of training reward models.** As we discuss in lines 041-043, due to different alignment purposes, multiple RMs are required. For example, PKU SafeRLHF trains 2 RMs and uses them for safety alignment.
>
>
> ## 4. When the quality of model-generated responses is actually higher than the reference answers, similarity-based scoring metrics can have counterproductive effects.
> ---
> - *This is a good question, and we believe it is also a common problem for traditional RLHF with RMs and RefAlign.*
> - We can also ask, **"When the quality of model-generated responses is actually higher than the chosen responses, what will happen to the traditional RLHF pipeline?"** Will the RM give a higher score than the chosen response? We think that the answer is NO. As researchers spend a lot of time and money to construct good preference data. For example, Llama 3[3] will ask humans to edit the chosen response and make it better.
> - **It is not accurate to simply say "similarity-based scoring metrics can have counterproductive effects".** Back to Eq.(5), if this high-quality response also gets the highest score in the sampled $K$ responses, this high-quality response will also be encouraged. For example, for safety alignment in Table 2, RefAlign with relatively low-quality reference from PKU-SafeRLHF is also better than the baseline.
>
> ## Reference
> ---
>
> [1] Anthropic. Training a Helpful and Harmless Assistant with Reinforcement Learning from Human Feedback. 2022.
>
> [2] Meta. Llama 2: Open Foundation and Fine-Tuned Chat Models. 2023.
>
> [3] Meta. The Llama 3 Herd of Models.
>
> [4] Yu Meng, Mengzhou Xia, and Danqi Chen. Simpo: Simple preference optimization with a reference-free reward. In NeurIPS, 2024.
>
> ---
>
> Please do let us know if you have any other questions.
>
> Thank You.
>
> Best,
>
> Submission7493 Authors

---

> > ### Comment · Reviewer_Rw8W · 2025-11-26
> >
> > Thank you for the author's response, which has addressed most of my concerns. However, regarding the fourth point, I disagree with your proposed reward model. Essentially, it treats the reference answer as an absolute standard. During the online generation phase, the model's response only receives a high score when it is similarity to the reference answer. In reality, there can be multiple ways to answer a single question, and the same question might have completely different approaches to answering it. This could lead to a loss in the diversity of the model's responses. Moreover, even if an incorrect answer may have a high similarity to the reference answer—for example, in the case of a math problem where the derivation process is correct but the final calculation is incorrect—this approach would also fail in such scenarios.

---

> > > ### Author Response · Authors · 2025-11-26
> > > **Position: A Supplement to the Reward Function, Not a Full Replacement for RMs**
> > >
> > > Hi Reviewer Rw8W,
> > >
> > > Thanks for your response.
> > >
> > >
> > > ## 1. there can be multiple ways to answer a single question
> > > ---
> > > - It is true that there can be multiple ways to answer a single question. This is also the case for safety alignment; there are two reference answers for the question, a helpful one and a harmless one. During alignment, we calculate the similarities for the two reference answers and use their combination as the final reward score.
> > > - As the above stated, a simple solution to this question is to provide multiple reference answers.
> > >
> > > ## 2. Moreover, even if an incorrect answer may have a high similarity to the reference answer—for example, in the case of a math problem where the derivation process is correct but the final calculation is incorrect
> > > ---
> > > - This is a good question, somehow, *it is not accurate*.
> > > - If we only consider the correctness of the answers like popular RLVR (without a process reward model), it is not accurate to say "an incorrect answer may have a high similarity to the reference answer". It is an incorrect answer, so it will get a low reward.
> > > - If we further consider the correctness of the derivation process (introduce process reward models), the calculation of rewards for the intermediate results and the final calculation is separate. The reference answer should include a reference derivation process and the correct answer. In this case, a correct derivation process will get a high score, and an incorrect final calculation will get a low score; the final reward is the combination of the two scores.
> > > - This question shows that such similarity-based rewards can potentially be a supplement to the process reward models. This is a little beyond the scope of this paper, considering that we conduct experiments for safety, confidence, and general preference alignment. We have some preliminary results, if the reviewer is interested, we can also post here.
> > >
> > >
> > > Thanks for your time,
> > >
> > > Best,
> > >
> > > Submission7493 Authors

---

### Official Review · Reviewer_2HJV · 2025-11-01

**Soundness:** 3
**Presentation:** 3
**Contribution:** 3
**Rating:** 6
**Confidence:** 2

**Summary:**

- RefAlign aligns language models to a provided reference answer using semantic similarity as the reward in a simple REINFORCE‑style update.
- It removes preference pairs and reward models, aiming for a compact pipeline with competitive results on safety alignment, confidence calibration, and general instruction benchmarks.
- The approach is practical and easy to implement, but performance depends on reference quality and careful length control.

**Strengths:**

- Originality: Unifies alignment around reference similarity and removes reward models and preference pairs.
- Quality: Strong results in safety and confidence alignment; competitive general alignment; reasonable ablations and comparisons.
- Clarity: Method and objective are easy to follow; training details are practical.
- Significance: Useful for bootstrapping smaller models from strong references, with lower data and engineering cost.

**Weaknesses:**

- Reference quality dependence: effectiveness hinges on high‑quality, diverse references; noisy or narrow sets can misalign models.
- Length bias and calibration: outputs tend to be longer; needs explicit length penalties or controls and more consistent calibration reporting (e.g., ECE).
- Judge consistency: results may vary by judge; add multi‑judge analysis and open‑source judge checks.
- Baseline breadth: include same‑compute, parallel comparisons to direct optimization baselines (DPO, SimPO, ORPO, KTO) with matched settings.
- Generalization: broaden evidence to more domains and harder tasks beyond the reported benchmarks.

**Questions:**

- How stable are results across different similarity metrics (e.g., BERTScore, BLEURT, cosine embeddings)?
- Do multiple references per prompt improve robustness, and how are conflicts resolved?
- What is the accuracy–length trade‑off under explicit length penalties or constraints, and how does calibration change?
- How do results differ when references are human‑written or curated versus model‑generated?
- Can you provide same‑compute, parallel comparisons to direct optimization baselines and report training stability?

---

> ### Author Response · Authors · 2025-11-20
> **Rebuttal**
>
> Hi Reviewer 2HJV,
>
> Thanks for your time and feedback. Below we will address your concerns point by point.
>
> ## 1. How stable are results across different similarity metrics?
> ---
> - For safety alignment in Table 2 and Figure 2, different similarity metrics including BERTScore, Meteor, and EmbedLlama (cosine similarity of Llama embeddings) achieve consistent improvements in harmlessness and helpfulness compared to the baseline. This shows that *RL with similarity-based rewards is quite robust across different metrics.*
>
>
> ## 2. Do multiple references per prompt improve robustness, and how are conflicts resolved?
> ---
>  - **Multiple references per prompt will not always improve robustness.** This is jointly determined by the quality of reference answers and the alignment purpose. If the reference answers are contradictory, we need to design special solutions according to the alignment purpose.
> - For example, in safety alignment in Table 2, each prompt has a helpful reference and a harmless reference. These two reference answers are contradictory in some cases[1]. In Eq.(8), we introduce a hyperparameter $ \alpha $ to control the importance of harmlessness for safety alignment. By adjusting $\alpha$, we can control the harmlessness of the aligned model, as shown in Table 2HJV-1.
> - **Table 2HJV-1.**  Comparison with Beaver-v3.0 with respect to **Harmlessness.** The referee is `gpt-4o`.
> |  Method  | $\alpha$ in Eq.(8) | Win   | Tie   |
> |:--------:|-----------------|-------|-------|
> | RefAlign | 3.0             | 45.12 | 21.95 |
> | RefAlign | 4.0             | 48.19 | 22.89 |
>
>
> ## 3. What is the accuracy–length trade‑off under explicit length penalties or constraints
> ---
> - Generally, if we increase the length penalties, the algorithm/model will try to hack the reward by producing shorter responses. During the optimization process, some longer responses with better quality will not be rewarded, and the model will tend to generate short responses. This will lead to a drop in accuracy after alignment. Thus, we need to set a proper length penalty hyperparameter, namely, seeking a proper trade-off between accuracy and length.
>
> ## 4. How do results differ when references are human‑written or curated versus model‑generated?
> ---
> - Currently, in all three tasks, including safety, confidence, and general preference alignment, no human-written answers are available [1,2,3]. We either use the chosen response in these datasets or mode-generated responses as the reference answers.
> - We can infer such differences from the existing results in Table 2, where we have two different reference sets: one is from PKU SafeRLHF[1] and the other is generated by Llama-3.3-70B-AWQ. As PKU SafeRLHF includes some low-quality annotations (Appendix C.1 in the paper), the reference set generated by Llama-3.3-70B-AWQ produces better results after safety alignment. This shows that we can improve the alignment results with better reference answers. *If we assume that human-written answers are better than model-generated answers, human-written answers should bring improvement to the aligned model.*
>
> ## 5. Baseline breadth: include same‑compute, parallel comparisons to direct optimization baselines (DPO, SimPO, ORPO, KTO)
> ---
> - *Please check the results of general preference alignment in Table 4, where we provide results of direct optimization baselines like DPO, ORPO, KTO, and ORPO.* With similarity-based rewards, RefAlign is comparable to these direct optimization baselines.
>
>
> ## Reference
> ---
> [1] https://huggingface.co/datasets/PKU-Alignment/PKU-SafeRLHF
>
> [2] https://huggingface.co/datasets/shuchangtao/CONQORD_dataset
>
> [3] https://huggingface.co/datasets/openbmb/UltraFeedback
>
> ---
>
> Please do let us know if you have any other questions.
>
> Thank You.
>
> Best,
>
> Submission7493 Authors

---

> > ### Comment · Reviewer_2HJV · 2025-11-25
> >
> > Thanks for the detailed response. I’ve already given a 6, so I’ll keep my score.

---

### Official Review · Reviewer_JyiU · 2025-11-02

**Soundness:** 2
**Presentation:** 3
**Contribution:** 2
**Rating:** 4
**Confidence:** 3

**Summary:**

This paper proposes RefAlign, a REINFORCE-style algorithm for language model alignment. The method uses the similarity between model generations and unary reference answers, measured by metrics like BERTScore, as a surrogate reward function. This approach aims to achieve alignment in various scenarios without requiring binary preference data or a separately trained reward model.

**Strengths:**

+ The paper is well-written and clearly organized, making the proposed method easy to understand.
+ The experiments and corresponding analysis cover a wide range of tasks. And the baseline methods are thorough.
+ The proposed framework is versatile and can be adapted to different alignment tasks, including safety and confidence alignment, by modifying the reward function.

**Weaknesses:**

+ The main problem is the issue of unfair comparison in the experimental setup. The authors use external data from Llama-3.3-70B-Instruct to generate reference answers for training RefAlign. It is not clear whether the baseline methods also utilized this external data. My concern is that RefAlign's performance gains may largely benefit from this additional information gain, rather than the algorithmic novelty alone.
+ The core mechanism of RefAlign appears to be a reinforcement learning version of distillation. Using a BERTScore reward to match outputs from a more powerful model is conceptually very close to distillation, which the paper also compares against. As the reward model optimizes for text similarity, the proposed algorithm may be useful to a part of the community, but the contribution remains limited.

**Questions:**

To my understanding RefAlign is a reinforcement learning algorithm, why not include other RL baselines (GRPO, PPO, DAPO...) ?

---

> ### Author Response · Authors · 2025-11-20
> **Similarity-based Rewards are low-cost alternatives to traditional RMs**
>
> Hi Reviewer JyiU,
>
> Thanks for your time and comments. Below, we will address your concerns point by point.
>
> ## 1. unfair comparison in the experimental setup. external data from Llama-3.3-70B-Instruct.
> ---
> - For confidence alignment in Table 3, we do not use Llama-3.3-70B-Instruct to generate reference answers. We directly use the chosen responses from the preference dataset as the reference answers. RefAlign is still comparable to baselines like CONQORD using PPO (with reward models) for confidence alignment.
>
> - For safety alignment and general preference alignment, **baselines utilize extra information from humans.** Safe RLHF recruits crowdworkers for data annotation (Appendix A in [1]). Baselines in Table 4 use PairRM [2] for preference data annotations, it is trained with Anthropic/hh-rlhf, openai/webgpt_comparisons, and openbmb/UltraFeedback. These preference data are annotated by humans or LLMs like GPT-4.
>
> - Overall, **RefAlign can use good chosen responses as reference answers, and further gain can be obtained via better reference answers from powerful models/humans.** Generating reference answers from powerful models is cheaper than recruiting crowdworkers for data annotation. We believe this is not a drawback but a merit of the method, especially for alignment of relatively small models.
>
> ## 2. a reinforcement learning version of distillation...the proposed algorithm may be useful to a part of the community, contribution limited.
> ---
> - Thanks for the recognition from the reviewer. It is true that the proposed method is kind of an RL version of distillation for cases where direct distillation is applicable, say general preference alignment in Table 4.
>
> - For safety alignment and confidence alignment, direct distillation is not applicable. There are two chosen responses for safety alignment, one harmless and one helpful. For confidence alignment, we need to align the dynamic confidence from the model during training. In these alignment cases, similarity-based reward function can be a cheap alternative choice besides reward models which require binary preference data.
>
> - **Overall, similarity-based reward function can be an effective low-cost supplementary reward function for alignment.** As we stated in the Abstract,
> >"When unary reference answers are available, such similarity-based rewards can circumvent the need for binary preference data and explicit reward modeling".
>
>   In all three alignment tasks including safety, confidence, and general preference alignment, we demonstrate that similairy-based reward function is a viable low-cost alternative to reward models. For cases where binary preference data is hard to collect or reward modeling is costly (lines 038~043), our method does provide another feasible alignment choice. We believe this is valuable for the community.
>
> - This question is also relevant to another broader topic: the difference between RL and SFT[3]. This needs further discussion.
>
> ## 3. why not include other RL baselines (GRPO, PPO, DAPO...)
> ---
> - **The main RL baseline compared in this work is PPO.** For safety alignment in Table 2, Beaver-v3.0 is the aligned model after 3 rounds of PPO with a reward model and a cost model. For confidence alignment in Table 3, CONQORD uses PPO to align the model. CONQORD trains an RM and uses it for the RL optimization pipeline.
>
> - In this work, we want to show that the similarity between sampled generations and reference answers can be a supplementary reward function for alignment. RefAlign is a very basic REINFORCE RL algorithm. It is the first alignment method that directly leverages similarity-based reward signals in a general RL optimization pipeline. Theoretically, other RL algorithms can also use similarity-based reward functions. *The change mainly happens in the reward functions.*
>
>
> ## Reference
> ---
> [1] Dai et al. Safe RLHF: Safe Reinforcement Learning from Human Feedback. ICLR 2024.
>
> [2] https://huggingface.co/llm-blender/PairRM
>
> [3] Chu et al. SFT Memorizes, RL Generalizes: A Comparative Study of Foundation Model Post-training. 2025.
>
>
> ---
>
> Please do let us know if you have any other questions.
>
> Thank You.
>
> Best,
>
> Submission7493 Authors

---

### Note · Authors · 2026-01-03

I have read and agree with the venue's withdrawal policy on behalf of myself and my co-authors.